# Clinical and Pathological Features of Flexural Deformities Associated with Myopathies in Foals [note 1]

**DOI:** 10.3390/vetsci12060557

**Published:** 2025-06-06

**Authors:** Maria Pia Pasolini, Luigi Auletta, Davide De Biase, Emanuela Vaccaro, Chiara Del Prete, Chiara Montano, Mariaelena de Chiara, Evaristo Di Napoli, Orlando Paciello, Giuseppe Piegari

**Affiliations:** 1Department of Veterinary Medicine and Animal Production, University of Napoli Federico II, 80137 Naples, Italy; emanuela.vaccaro@unina.it (E.V.); chiara.delprete@unina.it (C.D.P.); chiara.montano@unina.it (C.M.); mariaelena.dechiara@unina.it (M.d.C.); evaristo.dinapoli@unina.it (E.D.N.); paciello@unina.it (O.P.); giuseppe.piegari@unina.it (G.P.); 2Department of Veterinary Medicine and Animal Sciences (DIVAS), University of Milan, 26900 Lodi, Italy; luigi.auletta@unimi.it; 3Department of Pharmacy, University of Salerno, 84084 Fisciano, Italy; ddebiase@unisa.it

**Keywords:** equine, contractures, muscle disease, biopsy

## Abstract

Congenital flexural deformities (CFDs) are common in foals. The affected sites, severity of symptoms, and response to therapy vary from case to case. The etiopathogenesis is often unknown, and diagnosis is typically based on clinical examination, sometimes accompanied by radiological assessment. In contrast, children with congenital contractures require a complex diagnostic workup, which often includes a muscle biopsy. This study aimed to describe the diagnostic utility and findings of muscle biopsies in foals with flexural limb deformities. Muscle changes were observed in all assessed cases of CFDs. The most frequently identified muscle disorders included core-like myopathy, mild nonspecific myopathy, mitochondrial myopathy, polysaccharide storage myopathy, and congenital fiber-type disproportion. These findings suggest that muscle biopsy can provide valuable information for investigating congenital flexural deformities in equine medicine.

## 1. Introduction

Flexural Deformities (FDs) are common in foals and are characterized by a deviation in the caudo-cranial plane, resulting in persistent joint hyperflexion [1]. Congenital FDs (CFDs) are present at birth, while acquired ones develop during the growth period in young foals [2]. Various terms have been used to describe different clinical presentations, including limb contraction, tendon contracture, arthrogryposis, and contracted foal syndrome [3]. Limb contractions were observed in the forelimbs of 0.4% of all horses presented for examination at institutions participating in the Veterinary Medical Data Program [4]. Additionally, they were reported in 20% of necropsies performed on 608 deformed fetuses or newborn foals [5]. The etiology is largely speculative [1]. However, CFDs are likely to have a multifactorial etiology [6], with associations to in utero malpositioning, abnormal fetal development, malnutrition, or diseases in the mare [3,6], exposure to infectious and toxic agents during gestation, ingestion of locoweed or hybrid Sudan grass by pregnant mares, genetic defects, and hypothyroidism [3]. It was demonstrated that longer gestation length appears to reduce the odds of early-life flexural and angular limb deviations in Thoroughbred foals. However, the biological mechanisms behind this association have not been elucidated [7].

Some authors have reported neuromuscular disorders as one of the possible causes [1,8]. A case of Arthrogryposis Multiplex Congenita was reported [9]. Congenital FDs exhibit varying degrees of severity and may affect many limbs and joints [10,11]. Given the heterogeneity of causes, clinical presentations, and therapeutic responses, FD can be considered a symptom indicative of various underlying diseases [10]. Identifying the specific disease is crucial for selecting the appropriate therapy and determining the prognosis. In human medicine, congenital muscle weakness and contractures have been associated with numerous diseases [12,13], including disorders of the central and peripheral nervous system, the motor end plate, and muscle conditions that restrict fetal movement during development, which are believed to contribute to contractures [14]. Therefore, standardized early examinations, including serum creatine kinase (CK) testing, electromyography, nerve conduction velocity studies, neuroimaging, and muscle or nerve biopsy, are essential for assessing affected children [12,15]. In light of these observations, this study aimed (1) to investigate the clinical and pathological findings in congenital FD (CFD) cases in foals and (2) to retrospectively describe the abnormalities detected in muscle biopsies of foals affected by CFDs.

## 2. Materials and Methods

### 2.1. Study Design

A retrospective study of cases of FDs in foals, referred to the Department of Veterinary Medicine and Animal Production at the University of Naples Federico II, was performed over a 20-year period (2005 to 2024). Medical records of foals affected by CFDs, with or without hypotonia and/or muscle weakness, in which a muscle biopsy was performed, were evaluated. The foals came from the breeding farms monitored by veterinarians referring to the Department. The case selection criteria included (1) a clinical diagnosis of flexural deformities present at birth, (2) histologic diagnosis from muscle biopsy, and (3) a complete medical history and follow-up data. All available information was collected for both living and deceased foals. This included consanguinity, the presence of other contracted foals in the sire’s or mare’s offspring, assistance in standing and feeding after birth, and the timing of colostrum intake. The medical records were also evaluated to obtain information about the number and location of affected joints, the severity of the contracture, and the presence of other concurrent malformations. The results of routine bloodwork, including a complete blood count (CBC), electrophoresis of serum proteins, and a chemistry profile focusing on muscle markers such as CK, AST, and LDH, were also collected. When FD was associated with muscle weakness and mandibular prognathism, thyroid hormone activity was also evaluated. Finally, therapy and follow-up at one year were recorded for all assessed cases.

### 2.2. Muscle Biopsy Protocol

All cases evaluated in this retrospective study underwent muscle biopsy following a standard protocol. Biopsies were carried out on living foals with the owner’s consent or during necropsy on stillborn or deceased foals. Briefly, living neonate foals were sedated with diazepam (0.1 mg/kg, IV—Ziapam 5 mg/mL; Dechra Veterinary Products, Torino, Italy) and butorphanol tartrate (0.04 mg/kg, IV—Nargesic 1 mg/mL; ACME Srl, Corte Tegge Cavriago (RE), Italy). Local anesthesia was achieved with lidocaine hydrochloride 2% (Lidocaina 20 mg/mL; Ecuphar Italia Srl, Milano, Italy). After making a vertical incision in the skin and muscle fascia, two parallel incisions, 1 cm long and 0.5 cm apart, were made in the muscle. The muscle was grasped at a dorsal corner with forceps to prevent crushing other portions of the biopsy. A dorsal cross-sectional incision was then made, and the muscle sample was excised in a ventral direction. On stillborn or deceased foals, biopsies from muscles were collected within the first eight hours after death. Briefly, a 1 × 1 × 1 cm cube of muscle tissue was removed from the affected muscles. All samples were collected after a 1 cm skin incision. Samples were transported to the laboratory of the Department of Veterinary Medicine and Animal Production of the University of Naples Federico II within 5 min. Collected samples were frozen within 15 min using a standard protocol previously described [16].

### 2.3. Histological and Immunohistochemical Examination

Histological reports from muscle biopsies were collected to obtain information on the main histological and histochemical lesions, as well as on the final histological diagnosis. Overall, the histopathological reports included a description of the main morphological alterations observed with standard hematoxylin and eosin (HE) staining and a description of the main histochemical changes observed with periodic acid-Schiff (PAS), oil red-O, modified Gomori trichrome (Engel-Cunningham modification), reduced nicotinamide adenine dinucleotide tetrazolium reductase (NADH-TR), succinate dehydrogenase (SDH), cytochrome oxidase (COX), and adenosine triphosphatase (ATPase) staining [17]. When available, immunohistochemistry and electron microscopy results were also evaluated.

## 3. Results

Figure 1 presents a flow chart depicting the foals included in this study. Out of a total of 1623 foals born during the study period, 28 presented with CFD. Muscle biopsies were performed in 15 of these cases. The clinical records of fifteen foals were included in the study: 7 fillies and 8 colts. The breeds represented were 1 Appaloosa, 9 Standardbreds, 2 Warmbloods, 2 ponies, and 1 donkey. No clinical information about the sire’s progeny was available for any foals. The dam of one foal (No. 2) had previously delivered another foal with CFD, and her daughter also produced a contracted foal. Both cases were mild and resolved spontaneously with controlled exercise. The dam of the pony (No. 1), bred to the same stallion, had delivered a stillborn foal with CFD the previous year. No other familial correlations were found. Six foals (Nos. 9–14) experienced dystocic deliveries: two (Nos. 10 and 14) were born alive, while the other five (Nos. 9 and 11–13) were stillborn. One pony (No. 1), the donkey (No. 8), and the appaloosa foal (No. 15) were evaluated at 10, 12, and 3 months of age, respectively, with stiffness and FD present at birth, progressively worsening over time. Four foals (Nos. 4, 5, 7, and 8) required assistance to stand and maintain a standing position. In four cases (Nos. 4, 5, 6, and 7), colostrum intake was insufficient to provide adequate immunity. This was confirmed by serum protein electrophoresis in two cases (Nos. 5 and 7) and by a rapid SNAP^®^ Foal IgG Test (Idexx Laboratories, Hoofddorp, The Netherlands) in the other two (Nos. 4 and 6). Biopsies were collected in 6 live foals and during the necropsy in 9 cases. Patient information is summarized in Table 1.

### 3.1. Clinical Examination

In all foals, FDs were present at birth and were considered the cause of secondary complications, such as inadequate colostrum intake, septic complications, and rupture of the extensor tendons. In foals Nos. 1, 2, 5, 7–8, and 10, all parameters assessed during clinical evaluation were within the physiological range. Foal No. 3 was in critical condition and showed signs of respiratory distress. Two foals (Nos. 4 and 6) also exhibited patent urachus and omphalitis. The severity of FD and the affected joints are reported in Table 1. None of the foals that underwent necropsy had any additional gross malformations. Table 1 summarizes the sex, breed, age, and clinical symptoms of all assessed cases. Diagnosis, treatment, and follow-up have also been reported.

### 3.2. Hematochemistry

Muscle marker values ranged from normal to elevated. Foal No. 1 showed increased activity of AST, LDH, and CK. In foal No. 5, both LDH and CK were above the normal range, while in foal No. 7, only LDH was elevated.

In colt No. 7, which also presented with mandibular prognathism, thyroid hormone activity was altered. T4 levels (3.5 µg/dL) were within the reference range (2.9–5.25 µg/dL, mean 4.38 ± 0.38), while T3 (99.1 ng/dL; reference range 165–220 ng/dL), fT3 (2.6 ng/L; mean 13.3 ng/L), and fT4 (0.8 ng/dL; mean 1.7 ng/dL) were lower than the reported reference mean values [18].

### 3.3. Biopsy

Biopsies were performed on five living foals (Nos. 1, 2, 5, 7, and 10) and ten foals at necropsy (Nos. 3, 4, 6, 8, 9, and 11–15). In a pony (No. 1), the specimen was obtained from m. gluteus with the subject under general anesthesia. M. triceps brachii specimen was collected from three subjects (Nos. 2, 5, and 7): for one of them (No. 2), the procedure was carried out under general anesthesia, whereas the other two foals (Nos. 5 and 7) were sedated. Both m. gluteus and m. triceps brachii specimens were collected in foals, in which biopsy was performed post-mortem or under sedation (Nos. 3, 4, 6, and 8–15). The sampling sites were selected in the proximal limb muscles that were primarily affected by the deformity.

### 3.4. Histological Findings

Table 2 summarizes the histopathological alterations observed in study animals for each assessed case.

Case 1 and 14—COX-, SDH-, and NADH– stains showed well-delimited areas of reduced or absent oxidative activity localized in the center of the fibers. These areas showed circular shapes (cores) and were evident in both type-1 and type-2 fibers (Figure 2A,B). Furthermore, the cores appeared as light zones on PAS staining. Immunohistochemical examination using an anti-desmin antibody reported a desmin network markedly modified, with sharp delineation of the cores and abnormal distribution of the protein. On electron microscopy (EM), the core areas were characterized by preserved sarcomeres, with occasional myofibrils having slightly shorter length compared to the ones at the periphery of the fibers; the Z-lines were irregular and weaved or completely disrupted. Sarcoplasmic reticulum profiles and T-tubules were visible within the cores, which presented dilated and vacuolated cisternae and fewer mitochondria. Based on the morpho-histochemical changes, a definite diagnosis of Core-like myopathy (CCD) was made.

Case 2—Morphological and histochemical staining revealed the predominance of small type II fibers (Figure 2C,D). Based on this alteration, a final diagnosis of Congenital Fiber Type Disproportion (CFTD) myopathy was made.

*Cases 4, 6, 9, 11, and 13*—Histological examination of the muscle biopsies revealed only mild pathological changes, including moderate variability in fiber size and the presence of round, hypertrophic, and intensely stained degenerated fibers on both HE and Engel Trichrome stains. In one case (11), multifocal and polyphasic muscle fiber necrosis with sarcoclastosis was also observed. Therefore, the morphological diagnosis was consistent with mild, nonspecific myopathy.

Cases 5—H&E staining showed single atrophic fibers and fibers with a basophilic rim along the sarcolemma. Sections stained with Engel trichrome showed a high number of ‘‘red ragged fibers’’ with subsarcolemmal and intermyofibrillar deposits of reddish granular material (Figure 2E). SDH and NADH-TR stains reported a concentration of oxidative activity at the periphery of the fibers. Residual subsarcolemmal COX activity was observed in a few fibers (“moth-eaten fibres”) (Figure 2F). In this case, the ultrastructural examination was not performed. However, a predominance of type II fibers was found in case N. 5 on ATPase stain. Based on these findings, a diagnosis of mitochondrial myopathy (MM) was made in both assessed cases.

Case 7—Engel Trichrome staining reported several red intracytoplasmic aggregates consistent with nemaline-like rods. HE staining showed the presence of atrophic and degenerating fibers, while NADH-TR staining showed numerous fibers with a modified myofibrillar pattern. SDH staining reported an increase in enzyme activity in the subsarcolemmal location in many fibers (Pre-Ragged-Blue-Fibers). COX staining showed several moth-eaten fibers. Finally, ATPase staining at both pH (4.3–9.4) revealed a predominance of type II muscle fibers. Based on these findings, a diagnosis of nemaline-like myopathy was made.

*Case 8*—Morpho-histochemical staining revealed endo- and perimysial infiltration of adipose tissue. Changes in the number and size of fibers with centrally located nuclei were detected. Necrosis, fibrosis, and moderate and multifocal inflammatory infiltration were also reported. Based on these findings, a diagnosis of Lipomatous Myopathy (LM) was made (Figure 3A,B).

*Case 3*—Oil red O and ultrastructural examination showed intracytoplasmic and subsarcolemmal lipid droplets. Lipid accumulation was observed mainly, but not exclusively, in type 1 fibers. Based on this alteration, a final diagnosis of Lipid storage myopathy (LSM) was made (Figure 3C).

*Case 10*—HE and Engel’s Trichrome stains reported numerous pathological changes typical of neurogenic myopathy, mainly characterized by angulated atrophic fibers, often arranged in small groups; COX stain showed the presence of moth-eaten fibers; hyperchromatic fibers and hypertrophic neuromuscular junctions were also observed in the nonspecific Esterase stain. A second muscle biopsy, collected three months after the first, confirmed the presence of a neurogenic disorder. In this report, ATPase staining showed a predominance of type I fibers. These fibers appeared to be more severely affected than type II fibers, as commonly seen in various congenital myopathies [19].

*Case 12 and 15*—Histological description reported the presence of intrasarcoplasmic rimmed vacuoles, which appeared as clear spaces in both HE and Engel Trichrome stains. Most of these vacuoles were intensely pink-stained (PAS positive). Based on these observations, a final diagnosis of glycogen storage myopathy was made (Figure 3D).

Table 2 summarizes the main histological alterations and the final diagnosis for each assessed case.

#### Therapy and Follow-Up

Medical or surgical therapies were administered based on the severity of the deformity and the affected joint, following those reported in the literature. Treatments included intravenous oxytetracycline, casts or splinted bandages, pain management, corrective shoeing, and distal check ligament desmotomy, flexor tendons tenotomy [20,21,22]. The casts and bandages were applied with the foal under sedation or general anesthesia, extending from the radius to the hoof. Follow-up at one year is reported in Table 1.

## 4. Discussion

This study aims to shed light on the underlying cause of CFDs in horses, describing how primary and secondary myopathic alterations are consistently present in this condition. Until now, equine practitioners have focused mainly on treating the symptoms of CFDs, as the etiopathogenesis of the disease often remains unclear. This retrospective study includes a heterogeneous group of foals varying in breed, clinical presentation, hematochemistry, histopathology, response to therapy, and follow-up.

In human medicine, muscle weakness and contractures are the most reliable indicators of neuromuscular disorders and should be carefully assessed in infants with neonatal hypotonia [12,13]. Neuromuscular disorders and myopathies are among the possible etiology of CFD [3,23,24,25]. For example, muscle weakness, contractures, and stillbirth have been associated with congenital glycogen branching enzyme deficiency [26,27]. Some of the lesions observed in this study share several features with certain slowly or non-progressive neuromuscular disorders in humans—myopathies that are considered benign in children [27]. However, morphological similarity alone, in the absence of molecular investigations, may be insufficient for a definitive diagnosis.

To the author’s knowledge, no breed prevalence for CFDs has been recorded in the literature. In this retrospective study, of the 15 foals included, ten were Standardbreds; however, this apparent prevalence reflects the breed distribution in the study region. Nonetheless, epidemiological studies would be useful to determine the true prevalence of the disease across different breeds.

The prognosis for the study foals affected by CFD depended not only on the severity and etiology of the disease but also on the foal’s ability to stand and suckle colostrum as soon as possible after birth. Malformation and an inability to stand may lead to inadequate colostrum intake and failure of passive immunity [28]. Foals with FD should always be considered critical patients and treated in an intensive care unit (ICU) [29].

In the present study, CFDs more frequently affected multiple joints (mainly carpus, fetlock and interphalangeal joints), especially in both forelimbs. This finding is consistent with reports in the literature [3,5,10,14,21,30,31,32].

Muscle enzymes (CK, AST, and LDH) were elevated in foals affected by Core-like myopathy (No. 1), CFTD and Mitochondrial Myopathy (No. 5), and Myopathy with inclusion bodies (No. 7). High serum CK activity indicates acute muscle degeneration and LDH and AST activity in serum may also indicate muscle necrosis [27]. However, LDH and AST are not specific markers of muscle damage, as their elevations can also occur with liver necrosis. Therefore, muscle biochemistry profiles should also include ALT, GGT, and SDH activities to differentiate between muscle and liver necrosis [27]. Furthermore, in neonatal foals, interpreting elevated muscle enzyme activity in serum is challenging, as the increase may also be associated with trauma during parturition (dystocia), prolonged recumbency, weakness, as well as CFD [33,34].

The deformity completely resolved in cases Nos. 2, 5, and 7, after different therapies, including oxytetracycline, splint bandaging, and anterior extension of the affected limbs. Resolution of the deformities has been reported in a high percentage of affected foals, and spontaneous regression is also possible [3,20,35]. However, spontaneous or easy regression of the deformity, in the absence of other complications, may be due to the non-progressive nature of the underlying disease or to the postnatal maturation of muscles and nervous system. Moreover, soon after birth, flexor tendons and ALDDFT undergo further maturation, characterized by the regression of myofibroblasts, which are considered the target of oxytetracycline therapy [36]. In case No. 11, on the other hand, after an initial improvement with the application of rigid bandages and administration of oxytetracycline, clinical signs relapsed in the growing filly, likely due to the persistence of the underlying untreated disease. In foal No. 7, FD regression could also be related to the hormonal maturation. In foals, thyroid hormone concentrations are remarkably high at birth and decrease rapidly during the first few months of life [37]. Few papers report reference values for such hormones [38,39,40,41,42]. Hormone concentrations recorded in case No. 7 were lower than the normal values reported for foals of the same age. Hypothyroidism in foals might exert its negative effects during intrauterine life; indeed, although the normal equine placenta is quite impermeable to thyroxine, an increase in placental permeability could lead to considerable loss of foal’s T4, compromising the high levels necessary for foetal development [42]. Thus, developmental lesions in foals with hypothyroidism are often observed even when thyroid hormone concentrations are normal [37].

The interpretation of the mitochondrial deficit observed in the histoenzymatic stainings may be challenging. In human medicine, mitochondrial myopathies (MM) have been frequently diagnosed [18,43] and comprise a heterogeneous group of diseases characterized by defects in mitochondrial oxidative metabolism [44]. They may arise from mutations in either the mitochondrial or nuclear genome [25]. Changes in mitochondrial DNA sequences can be inherited or result from exogenous factors such as drugs or oxidative stress [18]. Clinical features can range from single-organ involvement to severe multisystem disease. Few cases of mitochondrial diseases have been described in veterinary medicine [45,46,47,48,49], and even fewer in horses due to limited testing. A deficit in mitochondrial activity was reported by Meijer and van den Hoven [50]), while Valberg et al. [51] described a case of a 3-year-old Arabian filly with extreme exercise intolerance associated with a deficiency of complex I respiratory chain enzyme (NADH CoQ reductase). Unfortunately, the limited number of cases currently reported in the literature makes it difficult to comprehensively assess the clinical features associated with the histopathology, as well as the mean survival time of horses affected by this heterogeneous group of diseases.

A diagnosis of congenital Fiber Type Disproportion (CFTD) was established in cases Nos. 2 and 5. CFTD is typically characterized by hypotonia and mild-to-severe generalized muscle weakness at birth or within the first year of life. The condition is slowly progressive, and only a few patients are able to walk in adulthood [51]. Contractures of the knees, hips, Achilles tendon, and spine are usually present at birth. Less commonly, the elbows and finger flexor or extensor muscles may also be affected [52]. The diagnosis of CFTD is based on a combination of clinical presentation and histological findings. Biopsies may reveal type 1 fibers that are at least 12% smaller than the mean diameter of type 2A and/or type 2B fibers in the absence of other significant pathological findings [51]. Pathological findings may change over time, allowing for refinement of the diagnosis through a second biopsy. In fact, some individuals initially diagnosed with CFTD may show a different condition on a second biopsy after some time [53].

It has been demonstrated that intra-uterine growth restriction (IUGR) might interfere with the maturation of both the muscle and nervous systems: IUGR reduces the formation of secondary myotubes, resulting in a decrease in the total number of myofibers and in muscle weakness at birth [54]. Normal muscle development depends on various hormones, including thyroid hormones, growth hormone, insulin-like growth factor, insulin, and steroid hormones. The complexity of the intrinsic and extrinsic regulatory mechanisms underlying muscle development and maturation highlights the need for further investigation into the impact of IUGR on muscle function [55]. It could be speculated that some cases of CFTD observed in muscle biopsies of contracted foals may be expressions of dysmaturity due to IUGR.

Cases Nos. 1, 3, 12, 14, and 15 involve primary myopathies. In foals Nos. 1 and 14, a complete set of analyses led to the diagnosis of a myopathy similar to human central-core disease (CCD), an inherited neuromuscular disorder characterized by central core-like lesions on muscle biopsy [56]. Typically, the disorder shows a predominance of type-1 muscle fibers, with few or absent type-2 fibers, and presents with single cores, often located at the center of the fiber [57]. In case No. 3, Lipid Storage Myopathy (LSM) was diagnosed. LSM is pathologically characterized by prominent lipid accumulation in muscle fibers due to lipid dysmetabolism [58]. In human medicine, making an accurate diagnosis through specific laboratory tests, including genetic analyses, is important, as some conditions are treatable [58]. In horses, similar genetic tests are still not available, and laboratory tests assessing free fatty acids, uric acid, ammonia, lactate, CK, glucose, and carnitine are expensive and typically reserved for research laboratories [59]. In cases Nos. 12 and 15, further studies would have been necessary to characterize the glycogenosis and to identify any genetic defects in the dam or sire.

In cases Nos. 4, 6, 9, 11, and 13, histology suggested an unspecific myopathy characterized by mild to moderate reduced or absent oxidative enzyme activity, and decreased adenosine triphosphatase (ATPase) activity was observed [60]. In cases with severe decrease or loss of enzymatic activity, immunohistochemical analysis for dystrophin proteins was performed and revealed normal expression of these molecules, and thus, a final differential diagnosis between congenital myopathic lesions and myonecrosis secondary to prolonged recumbency or trauma during dystocia was not feasible. Therefore, a final diagnosis of mild, nonspecific myopathy was made.

In case No. 8, a diagnosis of lipomatous myopathy (LM) associated with core-like lesions was made. LM is a degenerative muscle pathology characterized by the infiltration of adipose tissue into the muscle, occasionally reported in cattle, pigs, and rarely in horses, sheep, and dogs [61]. In Piedmontese cattle, LM has been associated with modifications of the COL18A1 gene and other genes affecting myogenesis and cell adhesion [62]. The biopsy of case No. 8 showed some common features with lesions described in Piedmontese cattle, such as fat infiltration of the muscle and an increase in the cross-sectional diameter of the fibers [62].

The owners of foals Nos. 5 and 7 declined to continue the investigation and repeat the muscle biopsy, so their records lacked data on their clinical and athletic follow-up. Studies on the long-term follow-up and athletic activity of horses affected by CFD are lacking, even though they could provide valuable insights into the disease.

## 5. Limitation

Because of its retrospective nature, this work has important limitations that need to be addressed. First, the absence of control groups may influence the correct interpretation of the results, making it difficult to demonstrate a direct correlation between assessed muscle histopathological alterations and the development of CFD in foals. Second, despite the long evaluation period, our study consisted of a small sample size. This was due to (1) the limited number of cases that underwent complete histopathological examinations and (2) the high percentage of Standardbreds in the study region. Indeed, the highest prevalence of CFD is observed in Thoroughbreds, although the literature lacks a comprehensive evaluation of the prevalence across different breeds. A low prevalence of CFD in Standardbreds compared to other breeds has been reported in a doctoral thesis [63]. In this context, the relatively low breed variability could be a further limitation of our study. Overall, our findings provide a reference basis for better investigating FDs in veterinary medicine. However, further studies will be needed to evaluate the prevalence of subclinical muscle abnormalities in foals without congenital flexural deformities, as well as to investigate CFD cases in larger sample sizes and populations with wider breed variability.

## 6. Conclusions

Several histopathological alterations were identified in muscle biopsies taken from contracted foals. Therefore, this retrospective study suggests that FDs should be regarded as a symptom of one or more underlying pathologies, and that each case should be approached individually with a tailored diagnostic process, including muscle biopsy when indicated. However, interpreting the described muscle changes and distinguishing between primary and secondary lesions can be challenging. Our data also suggest that neuromuscular disorders may be associated with both isolated and multiple contractures, potentially contributing to reduced fetal movements and intrauterine malpositioning, although further studies are needed to confirm this hypothesis.

## Figures and Tables

**Figure 1 vetsci-12-00557-f001:**
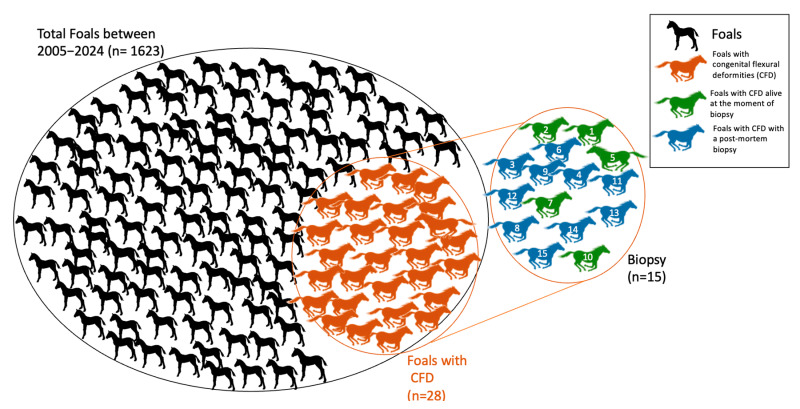
The flowchart of foals included in this study. Orange circle: foals diagnosed with congenital flexural deformities (CFD; *n* = 28). Right circle (Biopsy group): foals with CFD that underwent biopsy (*n* = 15). Each foal in the biopsy group is labeled with an individual identification number (1–15).

**Figure 2 vetsci-12-00557-f002:**
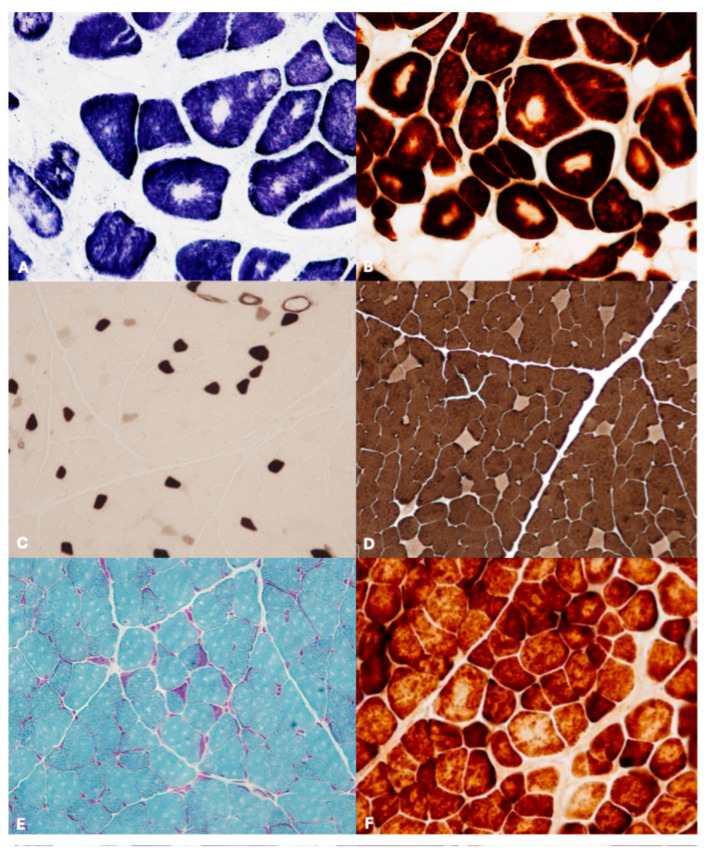
Representative histological alterations of study animals (**A**) Core-like myopathy (CCD) showing the absence of staining for cytochrome oxidase in the center of fibers (NADH staining; original magnification 40×); (**B**) COX stain with central cores lacking enzymatic activity (COX staining, original magnification 40×). (**C**,**D**) Congenital fiber type disproportion showing predominance of type II fibers stained light brown and dark brown with ATPase 4.3 and 9.4, respectively. ATPase staining, original magnification 20×). (**E**,**F**) Mitochondrial myopathy (MM) showing accumulations of subsarcolemmal reddish deposits consisting of mitochondria (ragged-red fibers) (Engle Trichrome staining, original magnification 20×) and irregular distribution of intermyofibrillar network (moth-eaten fibers) (COX staining; original magnification 20×).

**Figure 3 vetsci-12-00557-f003:**
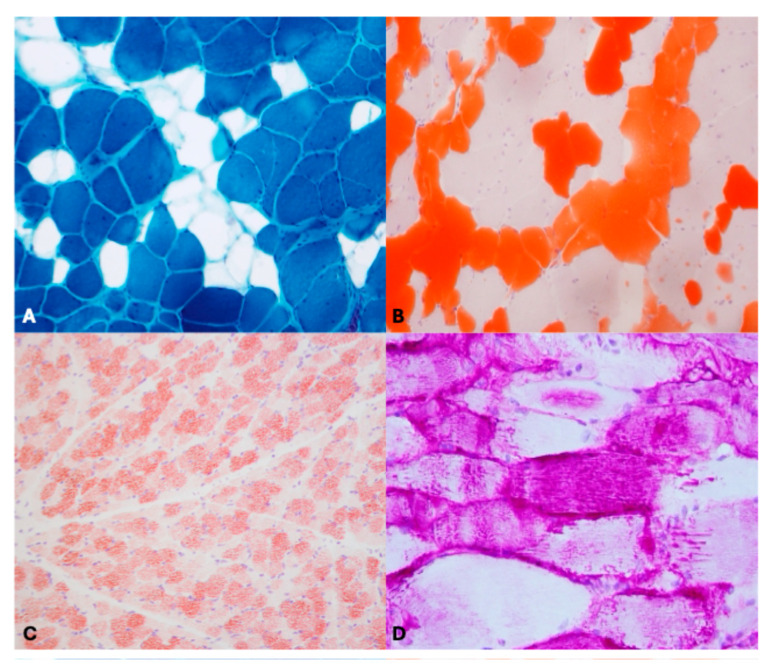
Representative histological alterations of study animals (**A**,**B**) Lipomatous Myopathy: Morpho-histochemical staining revealed endo- and perimysial infiltration of adipose tissue (Trichrome and Oil red O, original magnification 40× and 20×). (**C**) Lipid storage myopathy: intracytoplasmic and subsarcolemmal lipid droplets (Oil red O, original magnification 20×). (**D**) glycogen storage myopathy: accumulation of periodic acid Schiff (PAS)-positive material (Periodic acid Schiff, oral magnification 40×).

**Table 1 vetsci-12-00557-t001:** Sex, breed, age, clinical data, and contracture site recorded in the clinical dataset of the affected foals. M = male; F = female; SB = Standardbred; WB = Warmblood.

Case n.	Sex	Breed	Age	Symptoms	Site of Contractures	Therapy	Diagnosis	Follow-Up
1	M	Pony	10 months	Mild contractures, weakness	Metacarpo- and metatarso-phalangeal joints and interphalangeal joints of all limbs	None	Core like myopathy (CCD)	Death
2	F	WB	1 day	Mild contractures	Metacarpo-phalangeal joints of FL	Oxytetracycline IV, controlled exercise	Congenital Fiber Type Disproportion (CFTD)	Alive at one year after discharge
3	M	SB	1 day	Severe contractures, weakness, torticollis, scoliosis	Carpus, metacarpo/tarso- phalangeal and interphalangeal joints of all limbs	None	Lipid Storage Myopathy (LSM)	Euthanasia
4	M	SB	15 days	Mild contractures, weakness	Carpus, metacarpo-phalangeal of both FL	Cast, Oxytetracycline IV, Intensive therapy	Mild aspecific myopathy	Death
5	F	SB	1 day	Moderate contractures	Metacarpo-phalangeal, metatarso-phalangeal, and interphalangeal joints of the 4 limbs, more severe in FL	Cast, anterior extension, Oxytetracycline IV, controlled exercise	CFTD + Mitochondrial Myopathy (MM)	Alive at one year after discharge
6	M	SB	8 days	Moderate contractures, Weakness, Bilateral rupture of the common extensor tendon	Carpus and metacarpo-phalangeal joints of both FL	Cast, Oxytetracycline IV, Intensive therapy	Mild aspecific myopathy	Death
7	M	SB	1 day	Moderate contractures, weakness, mandibular prognathism	Carpus and metacarpo-phalangeal joints of both FL	Casts, Oxytetracycline IV, controlled exercise,	Myopathy with inclusion bodies	Alive at one year after discharge
8	M	Donkey	12 months	Severe contractures, bilateral patellar luxation	Metacarpo-phalangeal, metatarso-phalangeal, and interphalangeal joints of all limbs	Antinflammatory drugs	Core like myopathy, lipodisthrophy	Euthanasia
9	M	SB	0 days	Moderate contractures	Carpus and metacarpo-phalangeal of FL	None	Mild aspecific myopathy	Stillborn
10	F	WB	10 days 5 months	Moderate contractures, entropion of the inferior right eyelid	Carpus and metacarpo-phalangeal of FL	Casts, Oxytetracycline IV when neonate; distal check ligament desmotomy at the 5th month due to recurrence	Neurogenic myopathy (I and II)	Euthanasia
11	M	SB	0 days	Moderate contractures	Carpus and metacarpo-phalangeal of both FL	None	Mild aspecific myopathy	Stillborn
12	F	SB	0 days	Moderate contractures	Carpus and metacarpo-phalangeal of both FL	None	Polysaccharide Storage Myopathy (PSSM)	Stillborn
13	M	SB	0 days	Moderate contractures	Carpus and metacarpo-phalangeal of both FL	None	Mild aspecific myopathy	Stillborn
14	F	Pony	7 days	Moderate contractures	Carpus and phalangeal metacarpophalangeal of both front limbs	Casts, hyperimmune plasma, intensive therapy	Core-like myopathy (CCD)	Death
15	F	Appaloosa	3 months	Severe contractures	Carpus and metacarpo-phalangeal of both front limbs metacarpophalangeal of both front limbs	Casts, ozone therapy, and flexor tendons tenotomy	Polysaccharide Storage Myopathy (PSSM)	Death

**Table 2 vetsci-12-00557-t002:** Histopathological alterations observed in study animals for each assessed case; ”X” denotes that the item or criterion is present in the corresponding case.

Case	Atrophy	Central Nuclei	Necrosis	Degeneration	Inflammation	Fibrosis	Diagnosis
1	X	X	-	-	-	-	Core-like myopathy (CCD)
2	X	-	X	-	-	-	Congenital fiber type disproportion (CFTD) myopathy
3	X	-	-	X	-	-	Lipid storage myopathy (LSM)
4	-	-	-	X	-	-	Mild aspecific myopathy
5	X	-	-	-	-		CFTD + Mitochondrial Myopathy (MM)
6	-	-	-	X	-	-	Mild aspecific myopathy
7	X	-	-	X	X	-	Myopathy with inclusion bodies
8	X	X	X	X	X	x	Core-like myopathy lipodystrophy
9	-	-	-	X	-	-	Mild aspecific myopathy
10	X	-	-	-	-	-	Neurogenic myopathy (I and II)
11	X	-	X	X	-	-	Mild aspecific myopathy
12	X	-	-	-	-	-	Polysaccharide Storage Myopathy (PSSM)
13		-		X	-	-	Mild aspecific myopathy
14	X	-	X	-	X	x	Core-like myopathy (CCD)
15	X	X	-	-	X	x	Polysaccharide Storage Myopathy (PSSM)

## Data Availability

Clinical and pathological data are presented in this study and are available upon request from the corresponding author.

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
