# Peer review of "Clinical and Pathological Features of Flexural Deformities Associated with Myopathies in Foalsâ€"

_vetsci, 2025, doi:10.3390/vetsci12060557_

Round 1
Reviewer 1 Report
Comments and Suggestions for Authors
General Comments:
- The fact that no “control” animals were evaluated in this study should be emphasized and discussed further. Is the prevalence of subclinical muscle abnormalities in supposedly “normal” foals known? Without being able to compare CFD foals to normal foals, no conclusions can be drawn about the relationship between these muscle abnormalities and presence of CFD and instead labeling this as a descriptive study is more appropriate
- 16 foals is a very small case series for a 14-year period. Were there other CFD foals that presented without receiving muscle biopsy? Why did some foals receive muscle biopsy while others did not? Was this due to severity of disease, which may bias some of these results? And did the outcomes of foals without biopsy reflect those of the survivors in this study?
- The timing of biopsies, specifically of deceased foals or stillborn foals, should be discussed in more detail, as several foals were diagnosed with nonspecific myopathies that may have been attributed to timing of sampling, recumbency (in live foals), etc.
- It would be helpful for the relevant information to be present in a single table or paragraph in the discussion to make all this information easier for the reader to digest. This could include a table that has signalment, severity and site of contracture, histologic diagnosis, treatment, and outcome. Grouping the foals by their histopathology diagnosis might also be helpful for the reader to make conclusions, rather than always listing them in chronological order.
Specific Comments:
- Line 18: muscle changes were observed in all assessed cases of CFDs
- Line 39-41: please provide brief explanation for how this information aids in investigating flexural deformity; as you stated previously, most disorders are diagnosed via clinical exam and treated appropriately without the need for further diagnostics
- Line 58: correct this statement to say that some acquired limb contractures are associated with pain, as this is not always the pathogenesis
- Line 74: the second aim could be more specific to what was investigated, including investigating associations between presence of flexural deformities and abnormalities on muscle biopsy, as the biopsies did not change the diagnosis of “congenital flexural deformity” but did highlight that many of these foals also have changes to the muscles as well
- Line 78: how many foals total were seen in that period, and within that, how many fell within your criteria for inclusion? Did you collect muscle biopsies from any “control” foals (those without CFD)?
- Line 97-98: how long after death or stillbirth were muscle biopsies performed, and how was muscle necrosis after death controlled for?
- Line 133: inclusion of foals at 3, 10, and 12 months of age should be discussed further; was the contracture/deformity noted at birth and still present as they matured? When was the muscle biopsy taken and how often were these animals recumbent throughout their lives if they were larger and had significant contracture issues?
- Line 161: can you describe how/why each location was selected for biopsy?
- Line 252: can you include treatment in table 3 so the reader can more easily link the treatment with the disorder diagnosed and the ultimate outcome?
- Line 259: mislabeled, should be table 3
- Line 276: in children
- Line 288: “major” instead of mayor
- Line 303: can you summarize which muscle disorders were associated with these abnormalities instead of just referencing the case numbers? This would help the reader draw conclusions more easily
- Line 312: similar to above, can you summarize which muscle disorders were associated with resolution of signs and can you talk about which therapies were utilized?
- Line 326: although the normal equine placenta
- Line 339: Was this foal normal at birth, and if so, why was it included in a case series of CFDs?
- Line 398: this should be discussed more – can you specifically speak to the amount of time these foals were recumbent prior to sampling? Any previous studies to compare to for prolonged recumbency in foals or adult horses and presence of similar lesions”
- Line 413: it needs to be emphasized that without a control population, any link between CFD and muscle abnormalities cannot be made; this is a descriptive study only
Author Response
Reviewer 1:
General Comments:
- The fact that no “control” animals were evaluated in this study should be emphasized and discussed further. Is the prevalence of subclinical muscle abnormalities in supposedly “normal” foals known? Without being able to compare CFD foals to normal foals, no conclusions can be drawn about the relationship between these muscle abnormalities and presence of CFD and instead labeling this as a descriptive study is more appropriate
- 16 foals is a very small case series for a 14-year period. Were there other CFD foals that presented without receiving muscle biopsy? Why did some foals receive muscle biopsy while others did not? Was this due to severity of disease, which may bias some of these results? And did the outcomes of foals without biopsy reflect those of the survivors in this study?
- The timing of biopsies, specifically of deceased foals or stillborn foals, should be discussed in more detail, as several foals were diagnosed with nonspecific myopathies that may have been attributed to timing of sampling, recumbency (in live foals), etc.
- It would be helpful for the relevant information to be present in a single table or paragraph in the discussion to make all this information easier for the reader to digest. This could include a table that has signalment, severity and site of contracture, histologic diagnosis, treatment, and outcome. Grouping the foals by their histopathology diagnosis might also be helpful for the reader to make conclusions, rather than always listing them in chronological order.
AA: Dear Reviewer,
Thank you for all your comments. We have tried to address them and have worked to improve the experimental design by correcting the identified mistakes.
- To the best of our knowledge, the presence of subclinical muscle alterations in neonatal foals is unknown. We did not perform muscle biopsies in healthy foals or in foals that died from causes unrelated to CFDs. This and other limitations of the study have been highlighted in a paragraph added to the Discussion section.
- We agree that the number of biopsies retrospectively evaluated is relatively small in relation to the long duration of the study. Indeed, among the population of foals from the breeding farms that refer to our clinic, the prevalence of CFD is low. This is likely related to the predominant breeds in the region, such as the Italian Standardbred, for which the exact prevalence is unknown, but it is probably lower than in Thoroughbreds and Quarter Horses. Furthermore, the need to perform muscle sampling within a limited time frame and to obtain informed owner consent may have contributed to the small sample size. However, both severe and non-severe cases are represented in our cohort. A comment on this limitation has been added to the Discussion section. (lines 471-488)
- Thank you for your suggestion. The timing of the biopsies has been specified. Furthermore, we have discussed the potential influence of recumbency and delivery on nonspecific muscle alterations. (lines 446-453)
- The tables have been revised and reorganized for improved clarity.
- Specific Comments:
- Reviewer: Line 18: muscle changes were observed in all assessed cases of CFDs
- AA: thank you for your correction, “of” has been added (line 26)
- Reviewer: Line 39-41: please provide brief explanation for how this information aids in investigating flexural deformity; as you stated previously, most disorders are diagnosed via clinical exam and treated appropriately without the need for further diagnostics
- AA: Thank you for pointing that out. A brief explanation has been added, and the sentences have been revised as follows: “Even though many cases were diagnosed through clinical examination and successfully treated, we hypothesize that different underlying etiologies may present with similar flexural symptoms. A better understanding of these underlying causes is therefore desirable. These findings suggest that histopathological analysis may be a valuable tool for investigating flexural deformities in foals, although further studies are needed to evaluate the significance of the observed alterations.” (Lines 50-57)
- Reviewer: Line 58: correct this statement to say that some acquired limb contractures are associated with pain, as this is not always the pathogenesis.
- AA: Thank you for pointing that out. However, the acquired case has been removed, along with any references to acquired deformities.
- Reviewer: Line 74: the second aim could be more specific to what was investigated, including investigating associations between presence of flexural deformities and abnormalities on muscle biopsy, as the biopsies did not change the diagnosis of “congenital flexural deformity” but did highlight that many of these foals also have changes to the muscles as well.
- AA: Thank you for your suggestion: the second aim was changed in: “to retrospectively describe the abnormalities detected in muscle biopsies of foals affected by CFDs” (lines 95-96).
- Reviewer: Line 78: how many foals total were seen in that period, and within that, how many fell within your criteria for inclusion? Did you collect muscle biopsies from any “control” foals (those without CFD)?
- AA: Thank you, we appreciate your input. We considered it useful to indicate the total group of foals from which the reported cases were derived, and the text has been revised as follows: “Medical records of foals affected by CFDs, with or without hypotonia and/or muscle weakness, in which a muscle biopsy was performed, were evaluated. The foals came from the breeding farms monitored by veterinarians referring to the department.” (Materials and methods lines 101-104) ……..”Out of a total of 1,623 foals born during the study period, 28 presented with CFD. Muscle biopsies were performed in 15 of these cases” (Results lines 152-153).
- Reviewer: Line 97-98: how long after death or stillbirth were muscle biopsies performed, and how was muscle necrosis after death controlled for?
- AA: We agree with the reviewer’s concerns. Muscle samples were collected within 8 hours of death. Collected samples were frozen within 15 minutes using a standard protocol previously described. This information was reported in the final version of the manuscript (lines 132-133)
- Reviewer: Line 133: inclusion of foals at 3, 10, and 12 months of age should be discussed further; was the contracture/deformity noted at birth and still present as they matured? When was the muscle biopsy taken and how often were these animals recumbent throughout their lives if they were larger and had significant contracture issues?
- AA: As indicated in the Results section, one pony (No. 1), the donkey (No. 8), and the Appaloosa foal (No. 15) were evaluated at 10, 12, and 3 months of age, respectively. Stiffness FD were present at birth in all three and progressively worsened over time. In the Discussion, we further elaborated on the potential role of muscle alterations resulting from recumbency or trauma induced by either eutocic or dystocic delivery. (lines 446-453)
- Reviewer: Line 161: can you describe how/why each location was selected for biopsy?
- AA: Thank you for pointing this out. We added the following sentence: “The sampling sites were selected in the proximal limb muscles that were primarily affected by the deformity” (lines 212-213).
- Reviewer: Line 252: can you include treatment in table 3 so the reader can more easily link the treatment with the disorder diagnosed and the ultimate outcome?
- AA.: We have reorganized the tables and incorporated all the requested information.
- Reviewer: Line 259: mislabeled, should be table 3
- AA: Many thanks for identifying that mistake. We have now addressed this by modifying the tables.
- Reviewer: Line 276: in children
- AA: Thank you for pointing that out, the term was changed in “children” (line 334)
- Line 288: “major” instead of mayor
- AA: Many thanks for identifying that mistake. However this sentence has been deleted.
- Reviewer: Line 303: can you summarize which muscle disorders were associated with these abnormalities instead of just referencing the case numbers? This would help the reader draw conclusions more easily
- AA: Thank you for your suggestion, the muscle disorders were reported as follows: “ Muscle enzymes (CK, AST, and LDH) were elevated in foals affected by Core-like myopathy (No. 1), CFTD and Mitochondrial Myopathy (No.5), and Myopathy with inclusion bodies (No. 7).” (Lines 351-353)
- Reviewer: Line 312: similar to above, can you summarize which muscle disorders were associated with resolution of signs and can you talk about which therapies were utilized?
- AA: Thank you for your suggestion. However, we believe that including excessive detail regarding the clinical treatment is not directly pertinent to the objectives of the study. Thus, we have added the information about treatment and follow-up to Table 1, which was modified and combined with Table 3. Please let us know if you believe this is sufficient.
- Reviewer: Line 326: although the normal equine placenta
- AA: Many thanks for identifying that mistake. We have now addressed it.
- Reviewer: Line 339: Was this foal normal at birth, and if so, why was it included in a case series of CFDs?
- AA: Thank you for pointing this out; case No. 7 has been excluded from the study, and the remaining cases have been renumbered.
- Reviewer: Line 398: this should be discussed more – can you specifically speak to the amount of time these foals were recumbent prior to sampling? Any previous studies to compare to for prolonged recumbency in foals or adult horses and presence of similar lesions”
- AA: Thank you for your valuable comment: the topic was discussed more, and it was specified that in some cases “final differential diagnosis between congenital myopathic lesions and myonecrosis secondary to prolonged recumbency or trauma during dystocia was not feasible.” (lines 450-452).
- Reviewer: Line 413: it needs to be emphasized that without a control population, any link between CFD and muscle abnormalities cannot be made; this is a descriptive study only.
- AA: thank you for pointing that out; a paragraph about limitations of the study has been added (lines 471-489)
Reviewer 2 Report
Comments and Suggestions for Authors
The work effectively summarizes the diagnostic information on the pathology, which has found numerous specific insights over the years, with some interesting ideas.
The clinical case history may appear limited but offers ideas and comparisons with human medicine of a certain interest. In my opinion, this may represent a limitation of the work, but also its strength over time, since these works also have the task of maintaining high attention on a pathology of great impact in equine medicine.
Author Response
Dear Reviewer,
We sincerely appreciate your thoughtful feedback and are pleased that you valued our study.
Thank you very much.
Maria Pia Pasolini

Reviewer 3 Report
Comments and Suggestions for Authors
The topic of the article is interesting, but it is very poorly structured and has many errors in the experimental design.
Introduction
- Lines 57-58. The authors state that the neuromuscular etiopathogenesis has been demonstrated, citing a bibliographic review (1) and an exceptional case of Arthrogryposis Multiplex Congenita published in 1984 (7). This information is insufficient to justify the neuromuscular origin of such a common issue in equine practice. The authors should either remove this statement or find references that can support this claim.
- Lines 58-59. Pain is present in both congenital and acquired cases, and its presence does not allow for differentiation of the disease origin, as implied in that sentence. This is referenced in citation no. 8 (“Once a flexural deformity is present, an element of pain is involved and this pain can lead to a worsening of the deformity and/or an overload and resultant deformity of the contralateral limb”).
- In the objectives (line 74), only foals with congenital flexural deformity are considered. However, in the materials and methods section (line 127), foal number 7 with an acquired disease is included. That foal should be removed from the study since it does not align with the title or the inclusion criteria based on the objectives.
Materials and Methods
2.1. Study Design
Given the direct relationship between ischemic encephalopathy and the presence of flexural deformities, concomitant problems should be included. The same applies to septicemia or even hypoglycemia, which can cause foals to exhibit hypotonia or muscle weakness, clinically indistinguishable from a primary muscle problem. Since the authors have sufficient data, they should include in the tables the suspicion or confirmation of other diseases or problems.
- Results
- In general, the results are written in a very confusing manner, and data is mixed together. For example, lines 126-127 mention a total of 8 foals, whereas 16 are included in the study.
- Foal number 15 is included in the stillborn group in line 133; however, the table indicates that it was 7 days old.
- Foals 4, 5, 6, and 8 did not receive adequate colostrum, so their weakness and contracture could be due to this fact rather than a congenital problem. Additionally, foal number 6 had a rupture of the common digital extensor tendon, which could suggest the flexural deformity was acquired.
- I believe it is very risky to diagnose Core-Like Myopathy based solely on the clinical signs and histopathological data, without confirmation via a genetic test for the ryanodine receptor (RYR1).
- Likewise, it seems inconsistent that foals with mitochondrial myopathy survived past one year of age, considering the limited cases described in the literature usually indicate a poor prognosis.
There are many errors and disorganization of the data. I recommend rewriting this section and organizing the data into paragraphs. Including a flowchart could help clarify how many foals were finally included in the study, how many were alive, and how many were stillborn. Additionally, it would be helpful to know how many foals with acquired flexural deformities were admitted to the hospital during that period. Keep in mind that this is a very common issue in equine practice, and the prognosis is usually very good, so comparing the cases of both groups regarding prognosis and clinical findings would be valuable.
Discussion
- Line 269-270. Reference 12 is incorrect. Perhaps it is reference 14.
- The authors have focused the discussion on justifying congenital diseases when molecular techniques to confirm the diagnosis were not performed. The discussion should aim to differentiate foals with congenital conditions from those with acquired problems.
- There is no section or paragraph discussing the limitations of the study, despite the obvious presence of several. A limitations paragraph should be added before the conclusions.
Conclusions
The conclusions should be written more hypothetically, as there is no confirmation that these are indeed congenital myopathies. It should be considered that some animals were products of dystocia, so the histological findings might be due to inadequate gestational development rather than a true congenital issue.
References
Some references are very outdated, and important works related to flexural deformities are missing. For example:
Mouncey R, Arango-Sabogal JC, de Mestre AM, Verheyen K. Gestation Length is Associated With Early-Life Limb Deformities in Thoroughbred Foals. J Equine Vet Sci. 2023 Oct;129:104896. doi: 10.1016/j.jevs.2023.104896. Epub 2023 Aug 2. PMID: 37541603.
Author Response
Reviewer 3:
The topic of the article is interesting, but it is very poorly structured and has many errors in the experimental design.
AA: We appreciate your input and have worked to improve the experimental design, correcting the identified mistakes.
Introduction
- Reviewer: Lines 57-58. The authors state that the neuromuscular etiopathogenesis has been demonstrated, citing a bibliographic review (1) and an exceptional case of Arthrogryposis Multiplex Congenita published in 1984 (7). This information is insufficient to justify the neuromuscular origin of such a common issue in equine practice. The authors should either remove this statement or find references that can support this claim.
- AA: thank you for your comment. However, we prefer to retain this possible group of causes due to both the authority of the review author (Auer, J.) and the fact that this etiology is well recognized in human medicine. In this context, “hypotonia, weakness, contractures, respiratory and feeding difficulties are among the most common presenting symptoms of neuromuscular disorders in the newborn period” (Haliloglu G., European Journal of Pediatric Neurology, 2022). The text was changed as follows: “Some authors have reported neuromuscular disorders as one of the possible causes [1,8]. A case of Arthrogryposis Multiplex Congenita was reported [9] ”.(lines 79-80 )
- Reviewer: Lines 58-59. Pain is present in both congenital and acquired cases, and its presence does not allow for differentiation of the disease origin, as implied in that sentence. This is referenced in citation no. 8 (“Once a flexural deformity is present, an element of pain is involved and this pain can lead to a worsening of the deformity and/or an overload and resultant deformity of the contralateral limb”).
- AA: This sentence, along with any reference to acquired deformities, was deleted in order to focus the study on congenital deformities.
- Reviewer: In the objectives (line 74), only foals with congenital flexural deformity are considered. However, in the materials and methods section (line 127), foal number 7 with an acquired disease is included. That foal should be removed from the study since it does not align with the title or the inclusion criteria based on the objectives.
- AA: Thank you for your valuable comment; case 7 has been removed.
Materials and Methods
2.1. Study Design
- Reviewer: Given the direct relationship between ischemic encephalopathy and the presence of flexural deformities, concomitant problems should be included. The same applies to septicemia or even hypoglycemia, which can cause foals to exhibit hypotonia or muscle weakness, clinically indistinguishable from a primary muscle problem. Since the authors have sufficient data, they should include in the tables the suspicion or confirmation of other diseases or problems.
- AA: Thank you for the interesting suggestion. We have reorganized the clinical information into a single table. We agree that the relationship between sepsis, neonatal encephalopathy, congenital flexural limb deformities, and other neonatal diseases—such as umbilical disorders and meconium retention—in foals is complex and requires careful evaluation. Thus, we emphasized in the limitation paragraph that the focus of the study was to describe the observed muscle alterations and that further research is needed to clarify their significance.
Results
- Reviewer: In general, the results are written in a very confusing manner, and data is mixed together. For example, lines 126-127 mention a total of 8 foals, whereas 16 are included in the study. Foal number 15 is included in the stillborn group in line 133; however, the table indicates that it was 7 days old.
- AA: Thank you for the comment. The mistakes have been corrected.
- Reviewer: Foals 4, 5, 6, and 8 did not receive adequate colostrum, so their weakness and contracture could be due to this fact rather than a congenital problem. Additionally, foal number 6 had a rupture of the common digital extensor tendon, which could suggest the flexural deformity was acquired.
- AA: thank you for your suggestion; however, in all reported cases, inadequate colostrum intake resulted from difficulty in standing and suckling, and FDs were present at birth. Furthermore, extensor tendon rupture was consistently observed as a secondary consequence of the contracture present at birth, according with Mokry A. et al. EVJ 2022, DOI: 10.1111/evj.13893 “Clinical and imaging findings, treatment details and outcomes in foals with extensor tendon rupture—A multicentre retrospective study” that reports: “ Rupture of the extensor tendons (ETR) at the dorsal aspect of the carpus has been described in newborn foals ……. Normally, this condition occurs shortly after birth or within the first three to four days of life. Tendon rupture can occur as a primary problem or may be associated with an underlying flexural limb deformity (i.e., contraction of the flexor tendons).” We specify this in the text “In all foals, FDs were present at birth and were considered the cause of secondary complications, such as inadequate colostrum intake, septic complications, and rupture of the extensor tendons. (lines 184-186).
- Reviewer: I believe it is very risky to diagnose Core-Like Myopathy based solely on the clinical signs and histopathological data, without confirmation via a genetic test for the ryanodine receptor (RYR1).
- AA: We agree with the reviewer’s concerns. In human medicine, mutations in the RYR1 gene on chromosome 19q13.1 have been reported as a cause of Central core disease (CCD) in many cases. However, gene analysis can often be complex if the defining feature is absent or in cases where dual pathology exists (Sewry CA, Muller C, Davis M, Dwyer JS, Dove J, Evans G, Schroder R, Furst D, Helliwell T, Laing N, Quinlivan RC: The spectrum of pathology in central core disease. Neuromuscul Disord 12:930–938, 2002). Therefore, a final diagnosis is usually based on the clinical phenotype and the pathological assessment of the muscle biopsy specimen. In veterinary medicine, a few cases of myopathy with core-like structures (Core-like myopathy) have been described, and little is known about the etiology of the pathology in foals. Therefore, pathological analysis is currently considered the technique of choice for making a definitive diagnosis of core-like myopathy in horses (Paciello O, Pasolini MP, Navas L, Russo V, Papparella S. Myopathy with central cores in a foal. Vet Pathol. 2006 Jul;43(4):579-83. doi: 10.1354/vp.43-4-579. PMID: 16847006).
- Reviewer: Likewise, it seems inconsistent that foals with mitochondrial myopathy survived past one year of age, considering the limited cases described in the literature usually indicate a poor prognosis.
- AA: We agree with the reviewer’s concerns. Mitochondrial myopathies comprise a heterogeneous group of diseases characterized by defects in mitochondrial oxidative metabolism (Bottoni, P.; Gionta, G.; Scatena, R. Remarks on Mitochondrial Myopathies. J. Mol. Sci. 2023, 24, 124. https://doi.org/10.3390/ijms24010124). Unfortunately, the limited number of cases currently reported in the literature makes it difficult to obtain a comprehensive assessment of the mean survival time of horses affected by this heterogeneous group of diseases. We have added this comment in the Discussion section. (lines 388-390)
- Reviewer: There are many errors and disorganization of the data. I recommend rewriting this section and organizing the data into paragraphs. Including a flowchart could help clarify how many foals were finally included in the study (15), how many were alive, and how many were stillborn. Additionally, it would be helpful to know how many foals with acquired flexural deformities were admitted to the hospital during that period. Keep in mind that this is a very common issue in equine practice, and the prognosis is usually very good, so comparing the cases of both groups regarding prognosis and clinical findings would be valuable.
- AA: thank you for the suggestion; a flowchart has been added (fig.1). A limitation of this retrospective study was the absence of a control group, as noted in the limitations paragraph added at the end of the Discussion section.
Discussion
- Reviewer: Line 269-270. Reference 12 is incorrect. Perhaps it is reference 14.
- AA: We apologize for the mistake; the reference number has been corrected, and a more recent reference has been added.
- Reviewer: The authors have focused the discussion on justifying congenital diseases when molecular techniques to confirm the diagnosis were not performed. The discussion should aim to differentiate foals with congenital conditions from those with acquired problems.
- AA: Thank you for your valuable comment. The discussion has been revised. We emphasized that the flexural deformities were present at birth, whereas the significance of the observed muscle changes requires further evaluation.
- Reviewer: There is no section or paragraph discussing the limitations of the study, despite the obvious presence of several. A limitations paragraph should be added before the conclusions.
- AA: We appreciate your suggestion; accordingly, a limitations paragraph has been included in the revised manuscript. (lines 471-488)
- Conclusions
Reviewer: The conclusions should be written more hypothetically, as there is no confirmation that these are indeed congenital myopathies. It should be considered that some animals were products of dystocia, so the histological findings might be due to inadequate gestational development rather than a true congenital issue. - AA: the Conclusions have been written more hypothetically. (lines 491-500 )
- References
Reviewer: Some references are very outdated, and important works related to flexural deformities are missing. For example: Mouncey R, Arango-Sabogal JC, de Mestre AM, Verheyen K. Gestation Length is Associated With Early-Life Limb Deformities in Thoroughbred Foals. J Equine Vet Sci. 2023 Oct;129:104896. doi: 10.1016/j.jevs.2023.104896. Epub 2023 Aug 2. PMID: 37541603. - AA: we appreciate the suggestion and have added recent references.
Round 2
Reviewer 1 Report
Comments and Suggestions for Authors
Thank you for your thoughtful edits they significantly improved the quality of the paper.
The only specific comment was in Line 82 where it should read “joints” instead of “joint”
Author Response
Dear reviewer
Thank you for your valuable feedback. We are delighted that our study was well received.
- Reviewer: the only specific comment was in Line 82 where it should read “joints” instead of “joint”.
- AA: Thank you for your valuable comment. We apologize for the errors and have corrected them accordingly.
Reviewer 3 Report
Comments and Suggestions for Authors
The authors have done a great job revising the manuscript, and the article has reached a level of quality sufficient for publication.
Author Response
Dear Reviewer,
We thank you for your generous and constructive review. We're glad that you found value in our research.